# Synthesis of a Novel Linear α, ω-Di (Chloro Phosphoramide) Polydimethylsiloxane and Its Applications in Improving Flame-Retardant and Water-Repellent Properties of Cotton Fabrics

**DOI:** 10.3390/polym11111829

**Published:** 2019-11-07

**Authors:** Chaohong Dong, Ling Sun, Xingbo Ma, Zhou Lu, Pengshuang He, Ping Zhu

**Affiliations:** Institute of Functional Textiles and Advanced Materials, State Key Laboratory of Bio-Fibers and Eco-Textiles, College of Textile and Clothing, Qingdao University, Qingdao 266000, China; dongzhh11@163.com (C.D.); SunZero@yeah.net (L.S.); maxingbo11@126.com (X.M.); 18354263069@163.com (P.H.)

**Keywords:** flame retardant, water repellency, cotton fabrics, polysiloxane

## Abstract

A novel linear α, ω-di (chloro phosphoramide)-terminated polydimethylsiloxane (CPN-PDMS) was successfully synthesized and utilized as a formaldehyde-free water-repellent and flame-retardant for cotton fabrics. The flame retardancy of treated cotton fabrics was estimated by limiting oxygen index (LOI) test, vertical flammability test, and cone calorimetry test. The cotton fabrics treated with 350 g/L CPN-PDMS obtained excellent flame retardancy with an LOI value of 30.6% and the char length was only 4.3 cm. Combustion residues were studied using Fourier transform infrared spectroscopy (FTIR), scanning electron microscopy (SEM) and energy dispersive spectrometry (EDS) analysis. Results show that CPN-PDMS can effectively enhance water repellency and fire resistance of cotton fabrics. Furthermore, the breaking strength test and the whiteness test strongly prove that the tensile strength and whiteness of the treated cotton fabrics were slightly lower than that of the pure cotton fabrics. The wash stability test showed that after 30 laundering cycles, the treated cotton fabrics still had an LOI value of 28.5% and a water-repellent effect of grade 80, indicating that CPN-PDMS was an excellent washing durability additive. In summary, these property enhancements of treated cotton fabrics were attributed to the synergistic effect of silicon-phosphorus-nitrogen elements in CPN-PDMS.

## 1. Introduction

With the development of economy and improvement of people′s living standard, the function textiles have gradually been widely used in various industries and human life [1,2,3]. Cotton fabrics are one of the most important natural polymer materials, which possess superior properties such as softness, comfortability, breathability, and biocompatibility. Thus, cotton fabrics have acquired wide applications in the realm of clothing, house furnishing, industrial and military applications [4,5,6,7]. However, cotton fabrics have certain shortcomings such as high flammability, and poor hydrophobicity [1,8], which can easily lead to fire accident and limit its further application. Therefore, it is of great necessity to confer cotton fabrics with flame retardancy and water repellency.

In recent years, multifunctional cotton fabrics have been greatly developed, cotton fabrics with waterproofing, UV blocking, antibacterial resistance and flame-retardant have been reported by many studies [9,10,11,12]. In particular, in the past few decades, halogen-based and formaldehyde-based flame retardants have been widely used to improve the flame retardancy of cotton fabrics in terms of flame-retardant finishing of cotton fabrics [13,14]. However, with the enhancement of human environmental awareness and consideration of their own life safety [15,16], these flame retardants are gradually being phased out, and clean, harmless, and low-cost products have become an urgent demand of people.

To date, polysiloxane has attracted considerable attention in the area of fundamental research and industrial applications due to its unique properties, such as high thermal stability, biological compatibility, hydrophobicity, low toxicity, etc. It has been reported by research that Si–CH_3_ is an important hydrophobic group that can increase the water repellency of cotton fabrics [17]. Zhang et al. [18] synthesized a series of organosiloxane oligomeric (PDMS) with different chain lengths, and the hydrophobic properties were more obvious with the increase of PDMS chain length. At the same time, polysiloxanes have been promoted as the most promising new flame retardants for cotton fabrics because of their great thermal stability [19,20]. In order to further improve the flame retardant efficiency, some people also introduced phosphorus and nitrogen into organosilicon compounds [4,21], which constitutes a synergistic flame retardant system [22,23,24,25]. It is worth noting that almost no one will consider combining the flame retardancy and water repellency of polysiloxanes and applying them to cotton fabrics [26]. Therefore, the development of a multifunctional silicone-based auxiliaries for cotton fabrics has become a meaningful research work.

In this work, CPN-PDMS was synthesized as a novel agent with water repellency and flame retardancy, and it can be bound onto cotton fabrics by covalent bond due to its activity group. The water repellency and flame retardancy of treated cotton fabrics were evaluated by water contact angle, limiting oxygen index (LOI) and vertical flammability test, respectively. The combustion and thermal decomposition behaviors of untreated and treated cotton fabrics were investigated by cone calorimetry and thermogravimetry test. In addition, the surface morphologies and elemental composition of cotton fabrics before and after combustion were studied by SEM and EDS.

## 2. Materials and Methods

### 2.1. Materials

Scoured and bleached 100% plain-woven cotton fabrics (14.75 × 14.75 tex^2^, 122 g/m^2^) was purchased from Weifang Qirong Textiles Co., Ltd. (Weifang, China). Phosphorus oxychloride, allylamine, chloroplatinic acid, tetrahydrofuran (THF) were obtained from Tianjin FuYu Fine Chemical Co.,Ltd. (Tianjin, China). Ethanol, methanol, triethylamine, primary alcobol ethoxylate was applied from Sinopharm Chemical Reagent Co., Ltd. (Shanghai, China). Hydrogen silicone oil was supplied by Shandong Dayi Chemical Co., Ltd. (Qingdao, China). Toluene was obtained from Bodhi Chemical Co., Ltd. (Jinan, China).

### 2.2. Preparation of Ethyldichlorophosphate

In a 250 mL three-necked round bottom flask equipped with a stirrer and a condenser, phosphorus oxychloride (16.30 g, 0.10 mol) and 50 mL of THF (as solvent) were added. Ethanol (4.70 g, 0.10 mol) and 20 mL THF put into a separatory funnel, the solution was slowly dropped into the solution in the three-necked flask at 0–5 °C. After the dropwise addition was completed, the mixture was stirred at room temperature for 3 h under the protection of nitrogen. The products obtained after the completion of the reaction were subjected to vacuum distillation to remove the solvent and other volatile species. The corresponding synthesis process was given in Scheme 1.

### 2.3. Preparation of PDMS-2NH_2_


A mixture of hydrogen silicone oil (29.27 g, 0.04 mol), allylamine (5.52 g, 0.10 mol) and 50 mL toluene (as solvent) was added to a 250 mL three-necked round bottom flask charged with an argon inlet tube, a thermometer, and a reflux condenser. Then, the catalyst (0.7ml) was added to the mixed solution. The admixture was stirred at 90 °C for 10 h under nitrogen protection. After the reaction was completed, removal of the volatile material and purification of the product were performed by vacuum distillation to obtain a clear, yellow viscous compound (PDMS-2NH_2_, 83.1% yield). The specific synthesis process was shown in Scheme 2.

### 2.4. Preparation of CPN-PDMS

Dichloroethyl phosphate (16.30 g, 0.10 mol) and PDMS-2NH_2_ (42.11 g, 0.05 mol) were sequentially added to a 250 mL three-necked round bottom flask, and the mixture was placed in an ice bath and stirred 1 h. Then, the reaction was stirred at 35 ºC for 12 h under a nitrogen atmosphere. After the synthesis was completed, the resulting mixture was subjected to rotary evaporation to remove the solvent. The obtained oily compound (CPN-PDMS, 46.65 g, 85.2% yield) was used as a flame retardant and waterproof composite additive for cotton fabrics. The detailed synthesis process was shown by Scheme 3.

### 2.5. Preparation of Treated Cotton Fabrics

The cotton fabrics were soaked in finishing bath containing various amounts of CPN-PDMS, primary alcohol ethoxylate (6 g/L) and urea (50 g/L) with a bath ratio (*w*/*v*) of 1:10 at 25 °C for 60 min. After then, the samples after processing had 100% wet pickup after two dips and nips. Finally, the samples were dried at 80 °C for 3 min and cured at 150 °C for 4 min.

The weight gain (WG) of flame retardant added on cotton fabrics was calculated as follows:$$WG\;(\%)=\frac{W_{a}-W_{b}}{W_{b}}\times 100\%$$
where *W*_b_ and *W*_a_ represent the weights of cotton fabrics before and after flame retardant treatment, respectively.

### 2.6. Measurements and Characterizations

The Fourier transform infrared spectroscopy (FTIR) of CPN-PDMS and carbon residues of untreated and treated cotton fabrics were acquired using a Nicolet iS 50 FTIR spectrometer (Thermo Fisher Scientific, Waltham, MA, USA) over the wavenumber range of 500–4000 cm^−1^ using the ATR method. 

The limited oxygen index (LOI) values of the cotton fabrics treated with CPN-PDMS were determined by a LFY-606B digital limiting oxygen index apparatus (Shandong Textile Science Research Institute, Qingdao, China) according to the GB/T 5454-1997 standard.

Vertical flammability test was performed on a LFY-601A vertical combustion tester (Shandong Textile Science Research Institute, Qingdao, China) in accordance with the GB/T 5455-2014 standard.

The Contact angle (CA) of CPN-PDMS-treated cotton fabrics were evaluated with using optical contact angle & interface meter SL200KL in accordance with AATCC Test Method 22.

Thermogravimetric (TG) analysis of samples were carried out on a thermogravimetric analyzer STA6000 (PerkinElmer, Waltham, MA, USA) in the temperature range from 35 to 800 °C with a heating rate of 10 °C/min in air atmosphere.

The combustion behaviors of untreated and treated cotton fabrics were investigated using a FTT-0007 cone calorimeter (Fire Testing Technology Ltd., Shanghai, China) under a heat flux of 30 kW/m^2^ according to ISO 5660.

The surface morphologies of samples were observed using JSM-6010LA SEM apparatus (Japan Electron Optics Laboratory Co., Ltd., Tokyo, Japan) at an accelerating voltage of 15 kV. The surface needed to be coated with conductive gold by a sputter coater before observing the microstructure.

The contents of phosphorus (P), silicon (Si), oxygen (O), carbon (C) and nitrogen (N) for the CPN-PDMS-treated cotton fabrics before and after combustion were acquired with using energy dispersive spectrometer (EDS) (JEOL-6300F).

The washing resistance of samples were analyzed by a soaping fastness device in accordance with the GB/T 8629-2001 standard. The treated cotton fabrics were washed five times in 2 g/L neutral detergent as one washing cycle. 

Breaking strength tests for the fabric samples were measured based on the GB/T 3923.1-2013 standard, using a HD026PC multifunctional electronic fabric strength meter (Hongda Experimental Instrument Co., Ltd., Nantong, China).

The whiteness index values of untreated and treated cotton fabrics were referred by the GB/T 17644-2008 standard method using a X-rite 8400 (X-Rite Co., Grand Rapids, Michigan, USA).

## 3. Results and Discussion

### 3.1. Characterization of CPN-PDMS 

The chemical structure of CPN-PDMS was studied by FTIR analysis. As shown in Figure 1, the characteristic peaks at 2964 and 1036 cm^−1^ were attributed to stretching vibrations of –CH_3_ and Si–O–Si in the polysiloxane, respectively [12]. At 3204 cm^−1^, the stretching vibration peak of N-H in the final products was obviously visible on the spectrum [27]. Additionally, the absorption peaks at 734, 1079, and 1287 cm^−1^ were assigned to stretching vibration of P–N, P–O, and P=O, respectively. At 551 cm^−1^, the characteristic peak of P–Cl appeared on the spectrum, indicating that a sufficient reaction has occurred between dichloroethyl phosphate and PDMS–2NH_2_. It is worth noting that the characteristic peaks of Si–H and C=C were not shown in the spectrum, indicating that hydrogen silicone oil and allylamine were sufficiently reacted during the synthesis process. Additionally, the ^1^H NMR and ^31^P NMR spectra of CPN-PDMS, as can be seen in Appendix A. All of these absorption peaks arising from characteristic groups proved that the target product of CPN-PDMS was synthesized successfully. 

### 3.2. Flame Retardant and Mechanical Properties

In order to compare the flame retardancy of untreated and treated cotton fabrics, the LOI test and vertical flammability test results of cotton fabrics treated with CPN-PDMS of different concentrations were shown in Figure 2 and Table 1. It was clearly found through experimental data that with the increase in the amount of CPN-PDMS, the LOI values of treated cotton fabrics increased from 18.0% to 31.1% as well as the length of carbon residue decreased from 30 to 3.9 cm. In addition, the after-flame time of all treated cotton fabrics was 0 s, and when the concentration of the flame retardant reaches 350 and 400 g/L, the after-glow time of the treated samples was also 0 s. These data strongly proved that CPN-PDMS causes effective flame retardancy and was sufficient to reduce the damage of cotton fabrics in fire.

It is well known that whiteness and breaking strength are two important indicators for evaluating the mechanical properties of cotton fabrics treated with additives [28]. As can be seen from Table 1, following the incorporation of CPN-PDMS, the whiteness and tensile strength in both the warp and weft of the treated cotton fabric were gradually decreased, which was due to CPN-PDMS combined with fibers by covalent bonds to change the original structure and crystallinity of cotton fabrics. However, these changes were extremely weak that the effect on whiteness and tensile strength of fabrics was not significant. According to these results, the cotton fabrics were preferably treated by 350 g/L CPN-PDMS.

### 3.3. Thermal Degradation Stability

The thermal degradation process of the untreated and treated cotton fabrics was investigated by TG analysis in air atmosphere. The resulting curves for TG (a) and DTG (b) were shown in Figure 3 and the corresponding data were recorded in Table 2. It can be seen that cotton fabrics have two distinct stages of mass loss. At the first stage of 282.4 to 369.3 °C with a 68.42% mass loss, it was likely that the decomposition of cellulose to generate volatile substances and/or the dehydration reaction formed aliphatic char. At the second stage of 369.3–493.2 °C, the cotton fibers were decomposed completely, leaving a carbon residue of only 1.69%. The results may be that the chars produced in stage I were converted to an aromatic form [29], which were further oxidized to CO_2_ and CO. For cotton fabrics treated with CPN-PDMS, the temperature scope of the fastest thermal degradation was from 171.2 to 281.6 °C, and the remaining residue was 49.36%. Moreover, a residue of 29.13% remained at 700 °C, which attested to the excellent fire resistance caused by CPN-PDMS. It may result from crosslinking of the cotton by the phosphorus and silicon containing groups were crosslinked with the cotton fabrics at 171.2–281.6 °C to promote the formation of char layer [30]. Therefore, the treated cotton fabrics displayed better flame retardancy and thermal stability than the pure cotton fabrics.

### 3.4. Combustion Behaviors

The combustion behavior of untreated and treated cotton fabrics with 350 g/L CPN-PDMS had been explored by cone calorimeter. The acquired data were summarized in Table 3 and plotted in Figure 4. As shown in Table 3, the time to ignition (TTI) of the treated sample is 32 s, while the untreated sample is 2 s, which means that the treated cotton fabrics are more difficult to ignite. The heat release rate (HRR) and total heat release (THR) are two most important performance parameters to characterizing fire intensity. The THR and HRR of treated cotton fabrics were much lower than that of untreated cotton fabric, the peak heat release rate (PHRR) of treated cotton fabrics was only maintained at 66.1 kW/m^2^, and the untreated cotton fabrics was as high as 118.9 kW/m^2^ (shown in Figure 4a,b and Table 3). Effective heat of combustion (EHC) is the ratio of the mass and heat loss measured at a certain time. The average EHC of the cotton fabric treated with the flame retardant was greatly reduced to 2.08 MJ/kg compared to the value (8.69 MJ/kg) of the cotton fabric (shown in Figure 4c and Table 3), which mean the reduction of heat released and inhibited the combustion of volatile gases. It is also worth noting that the carbon residue (28.6%) of the cotton fabrics treated with CPN-PDMS was significantly higher than that of the pure cotton fabrics (shown in Figure 4d). This may be due to the treated cotton fabrics formed a polyphosphoric acid compound during combustion, which facilitated the formation of the carbon layer [31]. In addition, the CO_2_/CO ratio of treated cotton fabrics decreased from 22.75 to 7.8 (shown in Table 3), indicating that CPN-PDMS greatly reduced the combustion efficiency of raw cotton. All of indicates demonstrate that the treated cotton fabrics showed a marked improvement in resistance to flame.

### 3.5. Water Repellency

As shown in Figure 5, the change in hydrophobicity was characterized by comparing the water contact angles of untreated and treated cotton fabrics. The water contact angle of the treated cotton fabric increased to 148.1° relative to the raw cotton, and the finished sample possessed an excellent water repellency rating of 90. As known, polysiloxane with low surface tension and low surface energy can provide favorable hydrophobic properties on the cotton fabric. Since CPN-PDMS contains Si–CH_3_ to form a large amount of hydrophobic films to reduce surface tension, the treated cotton fabrics have excellent super water repellency [32,33].

### 3.6. Surface morphology

In order to further characterized the divergence in the surface morphology of untreated and cotton fabrics treated with CPN-PDMS, the samples were thoroughly investigated by SEM. It can be seen that the surface morphologies of pure cotton fabrics were relatively rough and maintained a complete natural shape (shown in Figure 6a). On the contrary, the treated cotton fabric was smoother and a dense membrane could be seen on its surface (shown in Figure 6c), which indicated the CPN-PDMS had been grafted to the cotton fabrics successfully. Secondly, as can be seen from Figure 6b and d, the untreated cotton fabrics lost their original integrity after combustion, whereas the cotton fabrics treated with CPN-PDMS still maintained carbon layer with continuous and complete. It can be inferred that the treated cotton fabrics formed a protective layer during combustion which was composed of granular and flocculent material made of polyphosphoric acid, SiO_2_ and other non-combustible substances, and acted as an effective barrier to flammable substances and heat transfer. Among them, the flame retardant mechanism exists in the following aspects: polyphosphoric acid can promote the rapid dehydration of cotton fibers to form a carbon layer and retard the spread of flame [34]; non-combustible gases can dilute the concentration of combustibles, resulting in fiber expansion and bubble generation [35]; the silicon element in CPN-PDMS forms a large amount of SiO_2_ during combustion, which greatly accelerates the formation of the carbon layer [36]. According to these results, it is apparent that CPN-PDMS is sufficient to impart the flame retardancy on cotton fabrics.

### 3.7. Elemental Analysis

The elemental composition of untreated and treated cotton fabrics surface was determined by EDS. As can be seen from Figure 7a, the surface of raw cotton contained only two major elements, C and O. However, the treated cotton fabrics further added Si, P, and N elements on the primeval basis (shown in Figure 7b), which indicated that a perfect combination was created between the CPN-PDMS and the cotton fabrics. In addition, a series of changes in the elemental content of the treated samples after combustion were clearly observed. Compared with the treated cotton fabrics, the mass percentage concentrations of Si and P in the residual carbon after combustion increased from 3.05% and 5.19% to 8.91% and 12.15%, respectively, which was attributed to the fact that CPN-PDMS formed SiO_2_ matrix and phosphoric acid derivatives in the flame [35]. On the contrary, the C, O, and N elements generated a non-flammable gas during combustion and were released into the air, which caused the corresponding weight concentration in the residual carbon to drop to 34.49% (C), 35.43% (O), and 1.34% (N) (shown in Figure 7c), respectively. Moreover, these elements homogeneously distributed on the burned cotton fabrics without visible aggregate (shown in Figure 7d), demonstrating the carbon layer formed after combustion has a high degree of homogeneity, which is consistent with the results of SEM observation. In summary, it can be concluded that CPN-PDMS is a finishing agent that can react with cotton fabrics and has superb flame-retardant capability.

### 3.8. FTIR Analysis of Char Layers After Combustion

The chemical structures of untreated and treated cotton fabrics after burning in air were characterized by FTIR and the acquired spectra were shown in Figure 8. Although there were many resemblances between the two carbon residues, some new characteristic peaks can be noticed in the treated sample. In the residual spectrum of the treated cotton fabric, the peaks at 1258 and 798 cm^−1^ were attributed the stretching vibrations of P=O and P–O [37], respectively, thereby demonstrating the formation of phosphoric acid and polyphosphoric acid. The characteristic peak at 1016 cm^−1^ was due to the bending vibration of Si–O–Si [38], which confirmed the formation of SiO_2_ matrix in carbon layer. The silicon element could construct a protective layer of silicon dioxide during combustion, which inhibited smoke and isolated heat. These results convincingly indicate that CPN-PDMS can not only effectively protect cotton fabrics from flammable gases and heat during combustion, but also promote faster carbonization of cellulose.

### 3.9. Washing Durability

The LOI and water contact angles values of cotton fabrics treated with 350 g/L CPN-PDMS after washing were shown in Table 4 and Figure 9. The treated cotton fabrics matched 2 g/L neutral detergent. Then, the fabrics were washed five times (each time for 12 min) at 30 °C in a washing machine as one washing cycle. It can be seen that the LOI and water contact angle values of treated cotton fabrics gradually decreased with the increase of laundering cycles. After washing 30 times, although the water repellency grade, LOI, and water contact angle values of the treated sample was lowered to 80, 28.5% and 118.6°, respectively, the requirements for the flame retardancy and hydrophobicity standards of cotton fabrics were still met [39]. In short, the grafting of CPN-PDMS with cotton fabrics is relatively strong, and therefore, has superior washing durability.

## 4. Conclusions

A new reactive finishing agent CPN-PDMS containing Si, P, and N elements was synthesized, and it was used to improve the flame retardancy and water repellency of cotton fabrics. The chemical structure of CPN-PDMS was characterized by FTIR. The elemental composition and surface morphologies of cotton fabrics before and after combustion were studied by EDS and SEM, respectively. The LOI and vertical burning test were carried out, when the CPN-PDMS concentration was 350 g/L, the LOI increased from 18% to 30.6%, and both the after-flame time and after-glow time of treated sample disappeared. The thermal degradation ability and combustion behavior of treated and pure cotton fabrics was studied by the thermogravimetry and cone calorimeter. It was shown that the residue of treated cotton fabrics at 700 °C (29.13%) was higher than that of pure cotton fabrics (1.46%) and the average EHC, HRR, THR, mass loss and CO_2_/CO values of the treated fabrics were lower than the pure fabrics. It was speculated that SiO_2_ matrix and polyphosphoric acid derivative were produced during thermal decomposition of CPN-PDMS, which promoted the formation of carbon layer. In addition, the treated cotton fabrics had water repellent rating of 90, the water contact angle increased to 148.1°, and had excellent washing stability. All of results proved that the treated cotton fabrics has superb flame retardant and water resistance.

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
