# Peer review of "Synthesis of a Novel Linear α, ω-Di (Chloro Phosphoramide) Polydimethylsiloxane and Its Applications in Improving Flame-Retardant and Water-Repellent Properties of Cotton Fabrics"

_polymers, 2019, doi:10.3390/polym11111829_

Round 1
Reviewer 1 Report
Despite the fact that the undertaken research is interesting and important from the application point of view, the way of presentation of the results obtained prevents their publication in its present form in my opinion.
The biggest problem stems from that the structures of the obtained final product (CPN-PDMS), as well as mid product (PDMS-2NH2), have not been properly confirmed. They should be further characterized by 1H, 13C, 29Si, and 31P NMR spectroscopy and GPC chromatography to confirm their structures. There is no information about the amount of catalyst used for allylamine hydrosilylation. The Add-on (weight gain) values were not presented. The composition of the finishing bath used for the preparation of treated cotton fabrics should be given in detail. There is no description of the washing procedure applied.
Author Response
Reviewer #1:
The biggest problem stems from that the structures of the obtained final product (CPN-PDMS), as well as mid product (PDMS-2NH2), have not been properly confirmed. They should be further characterized by 1H, 13C, 29Si, and 31P NMR spectroscopy and GPC chromatography to confirm their structures. There is no information about the amount of catalyst used for allylamine hydrosilylation. The Add-on (weight gain) values were not presented. The composition of the finishing bath used for the preparation of treated cotton fabrics should be given in detail. There is no description of the washing procedure applied.
Responses:
Thanks for your help reviewing our article and we really appreciate your suggestions.
According to your suggestion, weperformed 1H and 31P NMR tests on the final product (CPN-PDMS). The results show that the compound has been successfully synthesized (The results are shown in the attached PDF). In addition, the number of the hydrogenated silicone oil (raw material) repeating unit is “m=8” used in the experiment, it can be inferred that the number of the final product repeating unit is also “m=8”, so that it is not necessary to carry out GPC chromatography to characterize CPN-PDMS. The amount of catalyst used for allylamine hydrosilylationis 0.7 ml, we have added this data to the article (line 85). According to your suggestion, the Add-on (weight gain) values havebeen added in article (Table 1). According to your suggestion, the composition of the finishing bath for processing cotton fabrics has been explained in the article(line 103-104). The cotton fabrics were soaked in finishing bath containing various amounts of CPN-PDMS, primary alcohol ethoxylate (6 g/L) and urea (50 g/L) with a bath ratio (w/v) of 1:10 at 25 ºC for 60 min. According to your suggestion, we have described the detailed washing procedure in the article (line 293-295). The treated cotton fabrics matched 2 g/L neutral detergent. Then, the fabrics were washed five times (each time for 12 minutes) at 30 ºC in a washing machine as one washing cycle. Supplementary Note: The number of the final product (CPN-PDMS) repeating unitis “m=8”, its relative molecular mass is calculated to be 1095 g/mol (1000
Reviewer 2 Report
This review contains proposals for improvement of the article
Please notice the use of the slash (/)
Original text/ corrected text
The Tittle
Please verify if the chemical name ‘chloro phosamidate)’ is correct, I did not found
the word “phosamidate” in the Web of Science
in line 301 there is chlorophosphate siloxane for this compound, is it correct?
line 16
Fourier transform infrared (FTIR)/ Fourier 16 transform infrared spectroscopy (FTIR)
Line 17
energy dispersive spectrometer (EDS) analysis / energy dispersive spectrometry (EDS) analysis
line 31 –
soft/ softness
comfort/comfortability
permeability/ breathability
line 69
primary alcobol ethoxylate were applied from / primary alcohol ethoxylate was applied from
Question: which alcohol ethoxylate? There are many primary alcohols.
line 70
Hydrogen silicone oil (put here the trade name) was supplied by Shandong Dayi Chemical Co. Was it Dimethyl silicone oil for textile 350(DY-201 )?
How do you know its molecular weight ? ( from data in line 83 it is 732 g/mol)
Question to 2.3. Preparation of PDMS-2NH2 – What was the yield of this synthesis?
Line 92
PDMS-2NH2 (42.11 g, 0.05 mol) Question : did you measure the content of NH2 groups for estimating the molecular weight of PDMS-2NH2?
Line 95
the resulting mixture was subjected to rotary evaporation by a rotary evaporator to remove the solvent / the resulting mixture was subjected to rotary
evaporation to remove the solvent.
Line 96
The obtained products (CPN-PDMS, 46.65 g, 85.2% yield) were a flame retardant and waterproof composite auxiliary for cotton fabrics / The obtained product (CPN-PDMS, 46.65 g, 85.2% yield) was used as a flame retardant and waterproof composite additive for cotton fabrics
Remark : in the text replace auxiliary with additive
Line 97
Question : what was the physical appearance of CPN-PDMS? Did you perform elemental analysis to support its structure shown in the scheme 3? It is usually done foe new compounds. FTIR does not deliver quantitative results.
Line 146
was obviously appeated on the spectrum / was obviously visible on the spectrum
Line 148
It was worth noting / It is worth noting
Line 158
with the exaltation / with the increase
Line 162
had effective flame retardancy / causes effective flame retardancy
line 188
which certificated the superior flame retardancy of CPN-PDMS / which attested to the excellent fire resistance caused by CPN-PDMS
line 188
It may be because the phosphorus and silicon containing groups were crosslinked with the cotton fabrics / It may result from crosslinking of the cotton by the phosphorus and silicon containing groups
line 207
which meant the reduction/ which mean the reduction
line 208
It was also worth noting / It is also worth noting
Line 214
treated cotton 213 fabrics have a prodigious exaltation in resistance to flame/ treated cotton fabrics showed a marked improvement in resistance to flame
line 232
were thoroughly research by SEM/ were thoroughly investigated by SEM.
Line 327
cotton fabric s lost /untreated cotton fabrics lost
line 247
CPN-PDMS is sufficient to improve the flame retardancy of cotton fabrics/ CPN-PDMS is sufficient to impart the flame retardancy on cotton fabrics
Line 302
it was used to improved the flame retardancy/ it was used to improve the flame retardancy
Author Response
Reviewer #2:
This review contains proposals for improvement of the article
Please notice the use of the slash (/)
Original text / corrected text
The Tittle
Please verify if the chemical name ‘chloro phosamidate)’ is correct, I did not found the word “phosamidate” in the Web of Science
Line 301
there is chlorophosphate siloxane for this compound, is it correct?
Line 16
Fourier transform infrared (FTIR) / Fourier transform infrared spectroscopy (FTIR)
Line 17
energy dispersive spectrometer (EDS) analysis / energy dispersive spectrometry (EDS) analysis
Line 31 –
soft/ softness; comfort/comfortability; permeability / breathability
Line 69
primary alcobol ethoxylate were applied from / primary alcohol ethoxylate was applied from
Question: which alcohol ethoxylate? There are many primary alcohols.
Line 70
Hydrogen silicone oil (put here the trade name) was supplied by Shandong Dayi Chemical Co. Was it Dimethyl silicone oil for textile 350(DY-201 )?
How do you know its molecular weight ? ( from data in line 83 it is 732 g/mol)
Question to 2.3. Preparation of PDMS-2NH2 – What was the yield of this synthesis?
Line 92
PDMS-2NH2 (42.11 g, 0.05 mol) Question: did you measure the content of NH2 groups for estimating the molecular weight of PDMS-2NH2?
Line 95
the resulting mixture was subjected to rotary evaporation by a rotary evaporator to remove the solvent / the resulting mixture was subjected to rotary evaporation to remove the solvent.
Line 96
The obtained products (CPN-PDMS, 46.65 g, 85.2% yield) were a flame retardant and waterproof composite auxiliary for cotton fabrics / The obtained product (CPN-PDMS, 46.65 g, 85.2% yield) was used as a flame retardant and waterproof composite additive for cotton fabrics
Remark : in the text replace auxiliary with additive
Line 97
Question : what was the physical appearance of CPN-PDMS? Did you perform elemental analysis to support its structure shown in the scheme 3? It is usually done foe new compounds. FTIR does not deliver quantitative results.
Line 146
was obviously appeated on the spectrum / was obviously visible on the spectrum
Line 148
It was worth noting / It is worth noting
Line 158
with the exaltation / with the increase
Line 162
had effective flame retardancy / causes effective flame retardancy
Line 188
which certificated the superior flame retardancy of CPN-PDMS / which attested to the excellent fire resistance caused by CPN-PDMS
Line 188
It may be because the phosphorus and silicon containing groups were crosslinked with the cotton fabrics / It may result from crosslinking of the cotton by the phosphorus and silicon containing groups
Line 207
which meant the reduction / which mean the reduction
Line 208
It was also worth noting / It is also worth noting
Line 214
treated cotton 213 fabrics have a prodigious exaltation in resistance to flame / treated cotton fabrics showed a marked improvement in resistance to flame
Line 232
were thoroughly research by SEM / were thoroughly investigated by SEM.
Line 327
cotton fabric s lost / untreated cotton fabrics lost
Line 2
CPN-PDMS is sufficient to improve the flame retardancy of cotton fabrics / CPN-PDMS is sufficient to impart the flame retardancy on cotton fabrics
Line 302
it was used to improved the flame retardancy / it was used to improve the flame retardancy
Responses:
Thanks for your help reviewing our article and we really appreciate your suggestions.
Comment: Please verify if the chemical name ‘chloro phosamidate)’ is correct, I did not found the word “phosamidate” in the Web of Science
Answer: According to your suggestion, we have replaced “chloro phosamidate” with “chloro phosphoramide” in the article.
Comment: there is chlorophosphate siloxane for this compound, is it correct?
Answer: According to your suggestion, we have removed the phrase "chlorophosphate siloxane" in the article.
Comment: which alcohol ethoxylate? There are many primary alcohols.
Answer: The primary alcobol ethoxylate is the commonly used penetrant JFC.
Comment: Hydrogen silicone oil (put here the trade name) was supplied by Shandong Dayi Chemical Co. Was it Dimethyl silicone oil for textile 350(DY-201 )?
Answer: The hydrogen silicone oil is not dimethyl silicone oil used for textile 350(DY-201 ).
Comment: How do you know its molecular weight ? ( from data in line 83 it is 732 g/mol).
Answer: The number of the hydrogenated silicone oil (raw material) repeating unit is “m=8” used in the experiment, it is calculated that its relative molecular mass is 732 g/mol.
Comment: Question to 2.3. Preparation of PDMS-2NH2–What was the yield of this synthesis?
Answer: The yield of PDMS-2NH2 is 83.1%, we have already added this data to the article (line 88).
Comment: PDMS-2NH2 (42.11 g, 0.05 mol) Question: did you measure the content of NH2 groups for estimating the molecular weight of PDMS-2NH2?
Answer: The number of the hydrogenated silicone oil (raw material) repeating unit is “m=8” used in the experiment, so the repeating unit "m" of the intermediate product (PDMS-2NH2) was also 8, it can be inferred that the relative molecular mass of PDMS-2NH2 was 842 g/mol.
Comment: Remark : in the text replace auxiliary with additive
Answer: According to your suggestion, we have replaced auxiliary with additive in the article (line 98).
Comment: what was the physical appearance of CPN-PDMS? Did you perform elemental analysis to support its structure shown in the scheme 3? It is usually done foe new compounds. FTIR does not deliver quantitative results.
Answer: The physical appearance of CPN-PDMS is a transparent oily compound. we performed 1H and 31P NMR tests on the final product (CPN-PDMS). The results show that the compound has been successfully synthesized (The results are shown in the attached PDF).
Comment: Please notice the use of the slash (/). Original text / corrected text.
Answer: According to your suggestion, we have replaced “original text” with “corrected text” in the article.

Reviewer 3 Report
Dear editor,
First of all, thank you for asking me the review of article.
This article entitled “Synthesis of a novel linear α, ω-di(chloro phosamidate) polydimethylsiloxane and its applications in improving flame-retardant and water-repellent properties of cotton fabric” provides the preparation of silicon-phosphorus-nitrogen containing flame retardant which can react with cotton fabric and its effect on flame retardant and water repellent properties. This article is systematically written with clear objectives. I have the following question and comments for revision consideration.
In the ‘characterization of CPN-PDMS’ section, the chemical structure of CPN-PDMS was shown in Figure 1. Herein, I wonder which peaks are attributed to P-Cl bond in the structure because the residual P-Cl can react with cotton fabric. I also suggest that the chemical structure of each synthesis step be characterized via H NMR or FTIR. In Figure 3, the unit of y-axis should be revised from “oC” to “%”. The mentioned values in the text and those illustrated in the table are different. While the carbon residue of untreated cotton fabric is 1.69% in line 184, that of untreated cotton fabric in Table 2 is 1.46%. Also, while the CO2/CO ratios of untreated and treated cotton fabric are 23.30 and 7.18 in line 212, respectively, those of untreated and treated cotton fabric in Table 3 are different with text. Therefore, the authors should check it. In Figure 3, the thermal degradation of treated cotton fabric starts at lower temperature than untreated cotton fabric, due to phosphorus compound which has low thermal stability. In cone calorimeter test, however, TTI of treated cotton fabric increases from 2s to 32s. Why is the difference in these results? In Figure 9, the title of x-axis should be revised from “cotton angle” to “contact angle”. I suggest that the authors modify the font size and section number consistently throughout. There are some errors in English throughout the manuscript as follows. Please check it out. In line 69, “alcobol” → “alcohol” In line 130, “Silicon (Si), Oxygen (O)” → “silicon (Si), oxygen (O)” In line 185, “CO2” → “CO2” In line 236, “Figure 6 and d” → “Figure 6c and d”
Author Response
Reviewer #3:
In the ‘characterization of CPN-PDMS’ section, the chemical structure of CPN-PDMS was shown in Figure 1. Herein, I wonder which peaks are attributed to P-Cl bond in the structure because the residual P-Cl can react with cotton fabric. I also suggest that the chemical structure of each synthesis step be characterized via H NMR or FTIR. In Figure 3, the unit of y-axis should be revised from “oC” to “%”. The mentioned values in the text and those illustrated in the table are different. While the carbon residue of untreated cotton fabric is 1.69% in line 184, that of untreated cotton fabric in Table 2 is 1.46%. Also, while the CO2/CO ratios of untreated and treated cotton fabric are 23.30 and 7.18 in line 212, respectively, those of untreated and treated cotton fabric in Table 3 are different with text. Therefore, the authors should check it. In Figure 3, the thermal degradation of treated cotton fabric starts at lower temperature than untreated cotton fabric, due to phosphorus compound which has low thermal stability. In cone calorimeter test, however, TTI of treated cotton fabric increases from 2s to 32s. Why is the difference in these results? In Figure 9, the title of x-axis should be revised from “cotton angle” to “contact angle”. I suggest that the authors modify the font size and section number consistently throughout. There are some errors in English throughout the manuscript as follows. Please check it out. In line 69, “alcobol” → “alcohol” In line 130, “Silicon (Si), Oxygen (O)” → “silicon (Si), oxygen (O)” In line 185, “CO2” → “CO2” In line 236, “Figure 6 and d” → “Figure 6c and d”
Responses:
Thanks for your help reviewing our article and we really appreciate your suggestions.
At 551 cm-1, the stretching vibration peak of P-Cl in the final products (CPN-PDMS) was obviously appeated on the spectrum. According to your suggestion, I have already added this data in the article(149-150). According to your suggestion, we performed 1H and 31P NMR tests on the final product (CPN-PDMS). The results show that the compound has been successfully synthesized (The results are shown in the attached PPT). According to your suggestion, wehave changed the y-axis in Figure 3 to “%” in the article (Figure 3). At the second stage of 369.3-493.2 ºC, the cotton fibers were decomposed substantially completely, leaving a carbon residue of only 1.69%. This “1.69%” indicates the amount of carbon residue at 493.2 oHowever, “1.46%” in the table indicates the amount of carbon residue at 700 oC. According to your suggestion, wecompared the article to the table and calibrated the values of CO2/CO. The thermogravimetric (TG) test is mainly to analyze the thermal degradation process of the samples, while the cone calorimetry test is to analyze the combustion behaviors of the samples.The initial thermal degradation temperature of the treated cotton fabric is reduced due to the fact that CPN-PDMS can decompose at lower temperatures to produce phosphoric acid and polyphosphoric acid compounds. These phosphorus-containing compounds can be esterified and crosslinked with fiber macromolecules of cotton fabrics, which promotes the formation of carbon layers. However, the thermal degradation process does not represent the combustion behaviors of the treated cotton fabrics. The time to ignition (TTI) means that combustibles are burned in air or oxygen and must reach the minimum temperature required for the substance to ignite. This minimum temperature is called the TTI of the substance. Because CPN-PDMS has excellent flame retardancy, the TTI of treated cotton fabrics has increased. According to your suggestion, wehave changed the y-axis in Figure 9to “contact angle” in the article. According to your suggestion, we have adjusted the font size and part number in the articleand modified the existing English errors.

Round 2
Reviewer 1 Report
I'm satisfied with the amendments introduced and can recommend the revised manuscript for publication in Polymers.